# Machine Learning for Touch Localization on an Ultrasonic Lamb Wave Touchscreen

**DOI:** 10.3390/s22093183

**Published:** 2022-04-21

**Authors:** Sahar Bahrami, Jérémy Moriot, Patrice Masson, François Grondin

**Affiliations:** 1Faculty of Engineering, Université de Sherbrooke, Sherbrooke, QC J1K 2R1, Canada; sahar.bahrami@usherbrooke.ca (S.B.); j.moriot@allwavestechnologies.com (J.M.); patrice.masson@usherbrooke.ca (P.M.); 2All Waves Technologies, Sherbrooke, QC J1N 0C8, Canada

**Keywords:** signal processing, machine earning, ultrasonic wave, touch technology, human–machine interface

## Abstract

Classification and regression employing a simple Deep Neural Network (DNN) are investigated to perform touch localization on a tactile surface using ultrasonic guided waves. A robotic finger first simulates the touch action and captures the data to train a model. The model is then validated with data from experiments conducted with human fingers. The localization root mean square errors (RMSE) in time and frequency domains are presented. The proposed method provides satisfactory localization results for most human–machine interactions, with a mean error of 0.47 cm and standard deviation of 0.18 cm and a computing time of 0.44 ms. The classification approach is also adapted to identify touches on an access control keypad layout, which leads to an accuracy of 97% with a computing time of 0.28 ms. These results demonstrate that DNN-based methods are a viable alternative to signal processing-based approaches for accurate and robust touch localization using ultrasonic guided waves.

## 1. Introduction

The popularity of tactile sensor technologies in daily life (e.g., cell phones, access keypads, smart screens) leads to an increase in the demand for low-cost robust touch interfaces. Amongst the existing technologies [1,2,3], tactile surfaces based on ultrasonic guided waves answer the need for converting non-planar surfaces (including plastic material and transparent glass) into high resolution and durable tactile sensing surfaces. Moreover, this technology supports multi-touch detection and can estimate the contact pressure [4,5]. Acoustic based tactile sensors exploit the Surface Acoustic Waves (SAW) [6,7], or guided or Lamb Waves (LW) [4,8,9,10,11,12,13]. SAWs travel over the surface [14], whereas LWs propagate throughout the thickness of the material. The SAW tactile surfaces are vulnerable to surface contaminants such as liquid and scratches, and can only offer limited multiple-finger commands. On the other hand, Lamb Wave Touchscreen (LWT) technology supports multi-touch, which makes it more suitable for interfacing with smart devices [4,5]. LWT technologies are either passive [8,11,15,16,17] or active [4,9,10,12,13,18,19,20,21]. In passive LWT, touch action is the source of Lamb waves which can be identified by piezoceramic sensors located on the plate. In active LWT, the Lamb waves are both generated and received by piezoceramic elements. Static actions are not taken into account in passive mode since they do not create the wave field, while the active mode can overcome this deficiency [5,21].

While LWT offers interesting properties for numerous applications, it remains challenging to perform accurate touch localization. Many localization techniques have been proposed to address this challenge [22], some of them better suited to address the additional challenge of designing low-cost embedded systems. Some imaging-based algorithms localize the tactile position using Time of Flight (ToF) [23,24,25,26] and Delay-and-Sum beamforming [27]. Other techniques utilize damage indices (DIs) [28], which rely on correlation coefficients [29,30] to measure the similarity between the baseline and the input signals. Alternatively, localization can also be achieved using data-based learning methods [13,20]. Quaegebeur et al. [4,19] also proposed an ultrasonic touch screen based on guided wave interaction with a contact impedance. This technology is used to implement the glass tactile surface in this work and supports a wide range of screen sizes and solid materials. In this configuration, piezoceramic discs are used as four receivers and one emitter, installed around the plate. The contact impedance induces reflections and modifies the ultrasonic guided wave field. A Generalized Cross-Correlation (GCC) algorithm computes the similarity between the measured signal and a set of recorded signals to estimate the touch position. However, this approach requires costly hardware to process the large dictionary of reference signals, as it involves a significant amount of memory access and floating point operations (flops).

Besides this, Artificial Neural Networks (ANNs) used in Artificial Intelligence (AI) are applied in Structural Health Monitoring (SHM) [31,32] for damage identification and localization. A Multi-Layer Perceptron (MLP) is implemented for damage localization and quantification using the damage index. The k-Nearest Neighbor (kNN) algorithm with a Principal Component Analysis (PCA) feature extraction method is also investigated in [33]. A Support Vector Machine (SVM) for damage classification and localization was studied in [34], with features from different domains (time and frequency). The accuracy of this method, however, remains sensitive to noise and degrades to 57% when the Signal-to-Noise Ratio (SNR) drops by 30 dB. An auto-encoder-based approach for acoustic emission sources localization is studied in [35]. The RMS localization errors are 38 mm and 48 mm (for two metallic panels), which represents an improvement in comparison with SVM and ANN (78 mm and 67 mm, respectively). A singular value decomposition (SVD-PHAT) was investigated in [36,37] to localize multiple sound sources. The proposed method offers localization performance similar to SRP-PHAT but considerably reduces the computational load. Deep learning-based approaches have also been applied in SHM. Many SHM studies exploit Convolution Neural Network (CNN)-based models [38,39,40,41,42,43,44]. Rautela et al. [38] obtained damage detection and localization accuracy of 99% using 1D CNN. In the signal pre-processing phase, they performed five operations: (1) band-pass filtering and visualization; (2) frequency preferencing; (3) signal augmentation with noise; (4) cross-statistical feature engineering; and (5) TOF.

Other approaches from AI, such as Machine learning (ML) and Deep learning (DL) methods, can determine the coordinates of a finger in contact with a surface. A study has demonstrated that CNN can identify left and right thumbs with an accuracy of over 92% on capacitive touchscreens [45]. Chang and Lee [21] proposed a deep machine learning algorithm of 2D CNN for Lamb wave localization on an ultrasonic touch screen using 48 actuator-sensor paths. The 16 piezoceramic transducers contain 4 transmitters and 12 receivers. The CNN input data are an image with the dimensions of 12 × 2048 pixels, including data points gathered by the receivers, and achieved a positioning accuracy of 95%. To improve the localization performance, Li [22] proposed an ML-based lamb waves scatterer localization method, called CNN-GCCA-SVR. They trained a CNN-GCCA (deep version of generalized canonical correlation analysis) to extract the features, then used a Support Vector Regression (SVR) model for the localization task. This algorithm provides precise prediction using only one actuator and two sensors, with the localization errors between 2 mm and 12 mm depending on the sensing configurations.

The development of ML and DL methods in relation with ultrasonic Lamb wave technologies in recent years, and the demonstrated performance of AI in localization as mentioned above, encourage further investigations on ML and DL algorithms to improve accuracy in touch localization. In this study, a simple Deep Neural Network (DNN) is proposed to localize a finger in contact with a surface, using classification and regression approaches. The proposed approach also shows that a neural network can be trained for a specific keyboard layout to perform classification with high accuracy. To the best of our knowledge, this is the first time that a simple fully connected neural network is used for a robust touch localization on a tactile surface when exploiting ultrasonic guided waves. This work is organized as follows: in Section 2, the hardware setup is described. The dataset is introduced in Section 3. In Section 4, the localization methods employing classification and regression are investigated. The results are discussed in Section 4.3. The applications of the methods including access control keypad (Section 5.1) and tracking touch by human fingers (Section 5.2) are introduced in Section 5, which is followed by a conclusion (Section 6).

## 2. Hardware Setup

Figure 1 shows the processing pipeline. An array of five piezoceramic elements soldered to a flexible Printed Circuit Board (PCB) is bonded to the touch surface with regular epoxy. A custom LabVIEW interface from National Instruments (NI) is used to (1) control an NI-9262 module that emits an ultrasound signal through a linear amplifier driving the emitting piezoceramic, (2) control an NI-9223 acquisition module to record ultrasound signals measured by the four receiving piezoceramics amplified by a custom preamplifier, and (3) control a collaborative Universal Robot UR5e having six degrees of freedom (DoFs), which touches a glass surface with an artificial silicon finger at a desired position.

During the acquisition process, an emission signal excites the emitting piezoceramic element, which converts it into a mechanical wave that propagates through the host structure. When a finger touches the structure, it modifies the surface mechanical impedance, inducing reflections of the mechanical waves. These reflected waves are measured by the receiving piezoceramics, which convert them into electrical signals.

The touch surface is a tempered glass plate (shown in Figure 2) with dimensions of 20×20 cm and a thickness of 5 mm. The four receivers and the emitter are installed at the bottom of the glass, and consist of piezoceramic discs with a diameter of 6 mm and a thickness of 2 mm. The Universal Robot UR5e is equipped with a silicon finger with a diameter of 5 mm. Silicon is a material of choice for this application, as its mechanical impedance is close to a human finger. The *x* and *y* coordinates of the contact, such as contact pressure, are controlled accurately by the UR5e and recorded by the computer, which provides a reliable baseline.

## 3. Dataset

The robotic arm produces 6404 contacts at random positions and pressure on the glass surface. We set the robot to acquire random contact positions over the touch glass. The experiment was done for a given amount of time until a sufficient amount of data is generated. This dataset is split into training (70%), validation (20%), and test set (10%) signals. For each touch, a linear chirp excitation signal of 1 ms duration from 50 kHz to 100 kHz is generated with a sampling frequency of 500 kHz. The receiver signals are acquired during 1 ms at a sample rate of 250 kHz. Figure 3 shows how the features are extracted in the time and frequency domains. In the time domain, the 250-sample signals acquired for each of the four receivers are simply concatenated. In the frequency domain, the real and imaginary parts in the frequency range between 50 kHz and 100 kHz are selected from the Fast Fourier Transform (FFT) after applying a Hann window, which leads to four vectors of 49 elements. These vectors are concatenated to create a feature vector of 392 elements.

## 4. Localization

Localization consists of predicting the horizontal and vertical coordinates of the finger on the touch surface. The model takes the input signal in the time or frequency domain, and predicts the 2D-coordinate with a discrete grid using a classification approach, or in a continuous space using a regression approach.

### 4.1. Classification Approach

A classification approach is first investigated to estimate the contact point location on the touch surface. The 20×20 cm^2^ surface is first divided into N×N zones (or classes) of sizes $\frac{20}{N}\times \frac{20}{N}$ cm^2^, as shown in Figure 4. When class (i,j) is activated (where i∈{1,…,N} stands for the row index and j∈{1,…,N} for the column index), the estimated position of the touch contact corresponds to the center of mass of the zone, denoted as $c_{i,j}$.

Table 1 shows the nine different configurations explored in this work. As the grid resolution increases, the center of mass of each class gets closer to the exact touch position. However, the classification task also becomes more challenging, which can reduce the localization accuracy when the wrong class is selected.

Figure 5 shows a four-stage fully connected neural network to perform classification based on the input vector in the time ($x\in R^{1000}$) or frequency ($x\in R^{392}$) domain. Each stage consists of a cascade of a batch normalization layer, a linear layer, and a rectified linear unit activation function. The first stage outputs the tensor $h_{1}\in R^{400}$, while the second, third, and fourth stages generate $h_{2}\in R^{300}$, $h_{3}\in R^{200}$ and $h_{4}\in R^{100}$, respectively. A training time, a dropout layer where neurons are zeroed with a probability of 0.3 ensures regularization and prevents overfitting. Finally, the classification stage then consists of a linear layer followed by a softmax layer, which produces a one-hot vector of $\hat{y}\in {\left[0,1\right]}^{N^{2}}$ elements, with all zone indices concatenated ($\hat{y}=\left[{\hat{y}}_{\left(1,1\right)},{\hat{y}}_{\left(1,2\right)},\ldots ,{\hat{y}}_{\left(i,j\right)},\ldots ,{\hat{y}}_{\left(N,N\right)}\right]$).

For each touch *k* in the training dataset made of *K* elements, a one-hot label $y_{k}\in {\left[0,1\right]}^{N^{2}}$ is generated and corresponds to the zone that includes the touch position $t_{k}\in \left[0,20\right]\times \left[0,20\right]$. The cross-entropy loss function is computed between the target $y_{k}$ and the prediction ${\hat{y}}_{k}=f\left(x_{k}|\theta \right)$, where f(·|θ) stands for a neural network with parameters θ, and $x_{k}$ is the measured signals in the time or frequency domains. The optimal parameters θ are obtained during training using the Adam optimizer. At test time, the neural network predicts the vector ${\hat{y}}_{m}$ for each *m* out of *M* data points, from which the indices ${\left(i,j\right)}^{*}={arg max}_{\left(i,j\right)}{\hat{y}}_{i,j}$ are obtained. The estimated touch position ${\hat{t}}_{m}\in \left[0,20\right]\times \left[0,20\right]$ then corresponds to the center of mass of this zone, denoted as ${\hat{t}}_{m}=c_{{\left(i,j\right)}^{*}}$.

### 4.2. Regression

The regression approach aims at estimating directly the touch contact position. A four-stage fully connected neural network similar to the one introduced for classification is proposed in Figure 6. The architecture is identical, except for the output stage that contains only a linear layer that outputs a tensor $\hat{y}\in R^{2}$, which holds the horizontal and vertical positions of the touch. For each touch *k* in the training dataset, the target $y_{k}\in R^{2}$ corresponds to the touch position $t_{k}\in \left[0,20\right]\times \left[0,20\right]$. The mean square error (MSE) loss is computed between the target $y_{k}$ and the prediction ${\hat{y}}_{k}=f\left(x_{k}|\theta \right)$, where f(·|θ) stands for the neural network with parameters θ, and $x_{k}$ is the measured signal in the time or frequency domain. The estimated touch position ${\hat{t}}_{m}\in \left[0,20\right]\times \left[0,20\right]$ then corresponds to the prediction ${\hat{y}}_{k}$.

### 4.3. Results

The classification and regression approaches are validated with the test dataset. The localization performance is evaluated by comparing the estimated touch position and the baseline, using the Root Mean Square Error (RMSE):(1)$$\mathrm{RMSE}=\sqrt{\frac{1}{M}\sum_{m=1}^{M}{∥{\hat{t}}_{m}-t_{m}∥}_{2}^{2}},$$
where ${∥\cdot ∥}_{2}$ stands for the $l^{2}$ norm, and *M* corresponds to the number of data points in the test dataset. Note that the RMSE metrics penalize equally the position error in the horizontal and vertical dimensions. Figure 7 shows the RMSE (cm) as a function of the grid resolution *N* when using classification, and the RMSE when performing regression. The RMSEs are presented in time (blue) and frequency (red) domains. A 2×2 grid provides a mean localization error of 2.18 cm (standard deviation: 0.85 cm) and 2.1 cm (standard deviation: 0.8 cm) with frequency and time domains features, respectively. The RMSE reduces considerably when the classification touch zones are increased from 4 to 100 grids, as the center of mass of each class converges to the exact touch position when the grid resolution increases. With a 10×10 grid, the classification accuracy of 90% is achieved, which leads to a mean localization error of 0.41 cm (standard deviation: 0.25 cm) and 0.4 cm (standard deviation: 0.24 cm) with frequency and time domain features, respectively. The regression approach provides the mean localization error of 0.44 cm (standard deviation: 0.2 cm) and 0.45 cm (standard deviation: 0.21 cm) with frequency and time domains features, respectively. The results demonstrate that the RMSEs in the frequency and time domains are in the same range.

## 5. Applications

The results so far aimed at measuring and comparing the localization accuracy with respect to the model architecture and grid resolution. However, in a real scenario, the localization accuracy depends on the interface layout, and it is therefore important to measure the performance of the proposed architecture in such a realistic configuration. Thus far, the data were collected using a robot, which aims to mimic a human finger, but slightly differs due to material difference and pressure variation. This section shows how the proposed method performs with a virtual keypad interface, and illustrates the localization accuracy when a human user draws a shape with his finger. The feature extraction in frequency domain will be taken into consideration since it reduces the number of input parameters (392) compared to the time domain approach (1000).

### 5.1. Touch Localization on a Virtual Keypad Interface

The classification model previously introduced can be adapted and used to detect the touch coordinates on a virtual access control keypad, such as the one shown in Figure 8. This approach is appealing as the neural network is optimized to detect the keys for a given layout, which maximizes the localization accuracy. The same model architecture and dataset as the one formerly used for classification is chosen, except only the first two-stage fully connected neural network are considered. The first stage outputs the tensor $h_{1}\in R^{100}$, while the second stage generates $h_{2}\in R^{50}$. The output stage generates a tensor with 13 dimensions, where the first 12 classes correspond to the keys (1, 2, 3, 4, 5, 6, 7, 8, 9, *, 0 and #), and the last class represents the zone surrounding the keys (L).

The neural network predicts a vector with 13 elements, and the index of the element with the maximum value corresponds to the selected class. The confusion matrix in Figure 9 shows the performance of the classifier. According to the classification report, the overall accuracy is 97% with a computing time of 0.28 ms, using CPU (Intel Core i5). In this type of application, it is critical to avoid misclassification of a pressed key with another key. On the other hand, misclassifying a pressed key for the layout zone is less critical as the touch is simply ignored by the interface. The matrix diagonal represents the correct predictions as the true and predicted locations match. The proposed method classifies the pressed keys with an accuracy of 97%. There is no misclassification of a pressed key with another key. However, the neural network confuses some keys with the layout zone (class L), which can be disregarded by the interface.

To provide a basis for comparison with potentially simpler approaches, the ML approach kNN (with k = 1) [33] has also been used to detect the touch coordinates for the access control keypad. Figure 10 shows the accuracy of the DNN and kNN approaches using 100%, 80%, 60%, 40%, and 20% of the training data set. For example, D-20 corresponds to 20% of the training data set. The accuracy drops from 97% (96%) to 89% (87%) when the size of data are decreased by 80% using DNN (kNN). Accuracy of the DNN method is always greater than the kNN considering different sizes of the dataset. This shows an improvement in test accuracy as the training set is enlarged.

Figure 11 compares the computing time between DNN and kNN approaches. These results indicate that the DNN approach is 11.67 times faster than the kNN approach. In the kNN approach, compressing the training set will reduce the time (by 2.58 ms) needed to search for the number of neighboring points, thus speeding up the process. On the other hand, this leads to decreased accuracy as shown in Figure 10. In DNN, expanding the data improves the accuracy without increasing the computational time. Thus, the DNN approach is more effective in terms of accuracy and computational time. These observations motivated the choice of the DNN approach over the kNN approach.

### 5.2. Localization of a Human Finger

The proposed architectures with classification (Figure 5) and regression (Figure 6) are also validated with a real human finger that draws a circle on the touch glass. Figure 12 and Figure 13 show the human finger localization with classification with 10 grid resolution (C-10) and regression approaches, respectively.

The blue dots show the true position of the finger and the red squares indicate the predicted positions. Figure 14 presents the localization RMSE using classification (C-5 and C-10) and regression.The regression approach provides the mean localization error of 0.47 cm, with the standard deviation of 0.18 cm, which leads to a minimum error of 0.29 cm, which is in the range of localization error in Figure 7. However, mean localization error is 0.69 cm, with the standard deviation of 0.29 cm using classification with 10 grid resolution (100 classes) and exceeds 1 cm (with the standard deviation of 0.54 cm) when the number of classes drops to 25. The computing time is 0.44 ms for each test sample. The small amount of training samples in each class enlarges the RMSE when using classification. Moreover, the localization error is more pronounced when the grid resolution reduces as the center of mass of each class diverges from the exact touch positions. This can explain the poor performance of C-5 and C-10 as shown in Figure 7. Regression overcomes these issues and also offers the best results.

## 6. Conclusions

Localization techniques with classification and regression have been investigated on a glass touch surface with ultrasonic Lamb waves technology. The five piezoceramic elements are used as one emitter and four receivers installed at the bottom of the surface. In this study, a simple fully connected neural network is proposed to perform touch localization on the glass plate. The aim is to reduce the computational complexity of the analytical imaging approaches associated with touch localization techniques. A robotic arm with a silicon finger simulates the touch action with random position and contact pressure to train the deep neural network. Frequency-domain features are selected as it reduces significantly the number of input parameters compared to a time domain approach. The proposed processing architecture is then validated with a human finger to localize the touches, which leads to a mean error of 0.47 cm and standard deviation of 0.18 cm. The computing time is 0.44 ms for each test sample. The classification approach is applied for touch detection on an access control keypad, which provides an accuracy of 97% with a computing time of 0.28 ms for each unseen example.

During the data acquisition, the human fingers can be swiped on the screen and the touch pressure can change. This sets a limit on the accuracy of a signal measurement when gathering human finger data. The difference between signals generated by the human finger and the artificial finger should be taken into account. The human finger and artificial finger localization errors (presented in Figure 7 and Figure 14, respectively) are in the same range. The results validate the similarity between the signals created by the simulated touch and those generated by the real touch. The performance of the classification for the touch localization is, however, limited by grid resolution. As the grid resolution decreases, the center of mass of each class moves away from exact touch position, decreasing the localization accuracy. As shown in Figure 7, increasing the grid resolution improves the localization accuracy for the classification approach. Under such circumstances, the regression remains valid to localize the touches.

This analysis demonstrates the viability of the regression with a four-stage fully connected neural network for touch localization on an ultrasonic Lamb wave touchscreen. The classification with a two-stage fully connected neural network is preferred for the touch zone detection due to its ability to provide high-precision touch detection on an access control keypad. The results indicate that a current analytical, and computationally intensive, touch localization algorithm with a simple fully connected layer is possible.

In future work, this could be extended to multi-touch scenarios. Multi-touch artificial fingers can be designed to acquire multi-touch signals. The double-touch simulator is shown in [5]. The other possibility would be training and testing the model with human fingers. This could provide more accurate results, but, in return, may require more time to collect the data. Moreover, the robustness of the model can be affected by the touch pressure of human fingers. In multi-touch gestures, it is critical that all the fingers will be pressed and released while taking data to avoid mislabeling of the number of touches. The proposed model in the current study is easily scalable to surfaces with ultrasonic Lamb waves technology of different shape, size and material, including not being limited to plastic and metal.

## Figures and Tables

**Figure 1 sensors-22-03183-f001:**
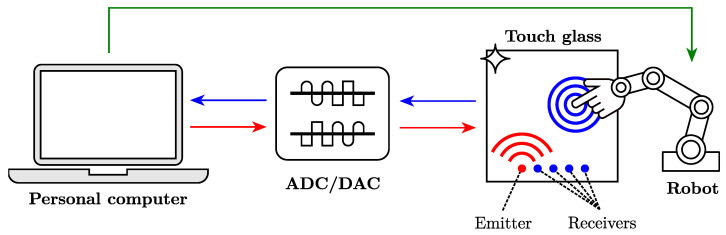
Schematic of the experimental setup.

**Figure 2 sensors-22-03183-f002:**
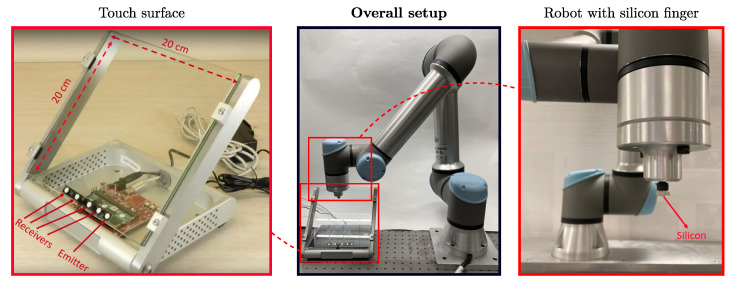
Hardware setup with the touch glass and the robotic arm.

**Figure 3 sensors-22-03183-f003:**
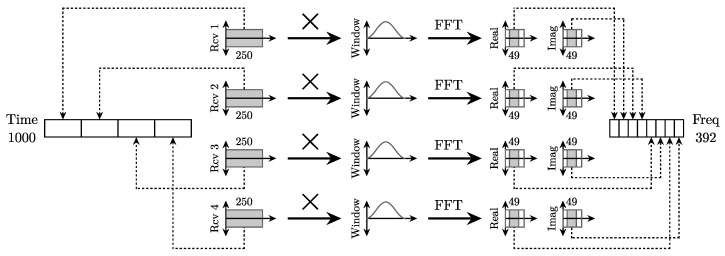
Features extraction in the time and frequency domains.

**Figure 4 sensors-22-03183-f004:**
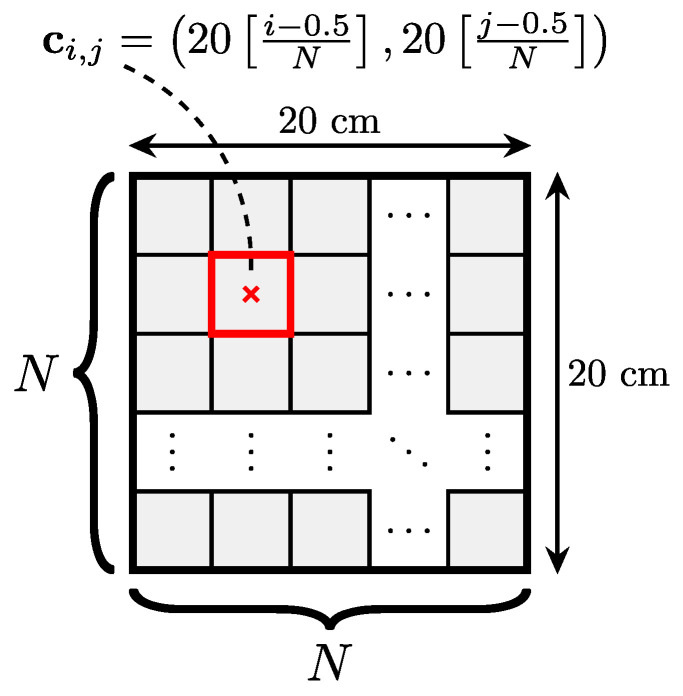
Classification touch zones. When zone (i,j) gets selected, the estimated touch position corresponds to the center of mass of this zone, denoted as $c_{i,j}$.

**Figure 5 sensors-22-03183-f005:**
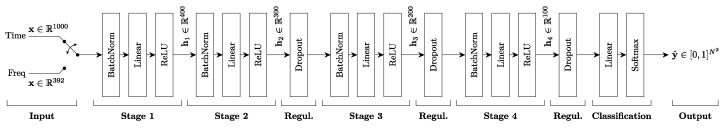
Neural network architecture for classification.

**Figure 6 sensors-22-03183-f006:**
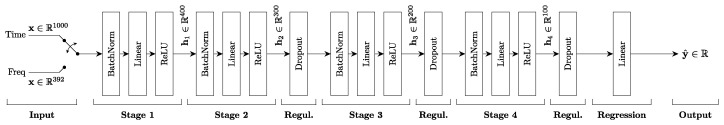
Neural network architecture for regression.

**Figure 7 sensors-22-03183-f007:**
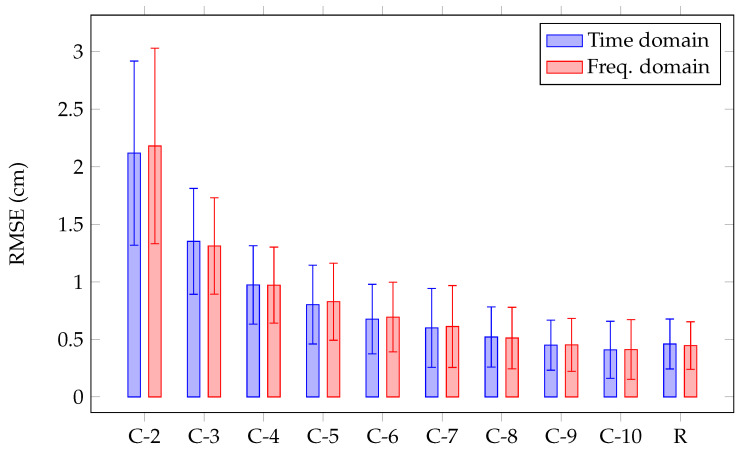
Comparison of localization RMSE (cm) between classification with grid resolution *N* (C-*N*), and regression (R) in frequency and time domains.

**Figure 8 sensors-22-03183-f008:**
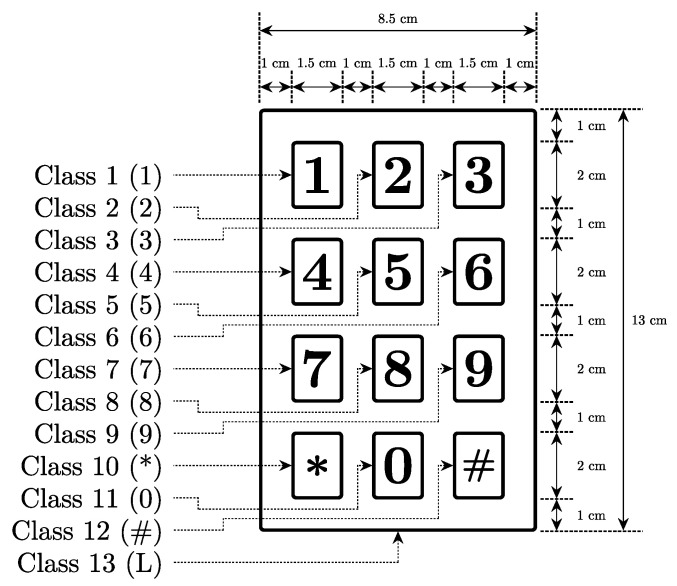
Access control keypad layout.

**Figure 9 sensors-22-03183-f009:**
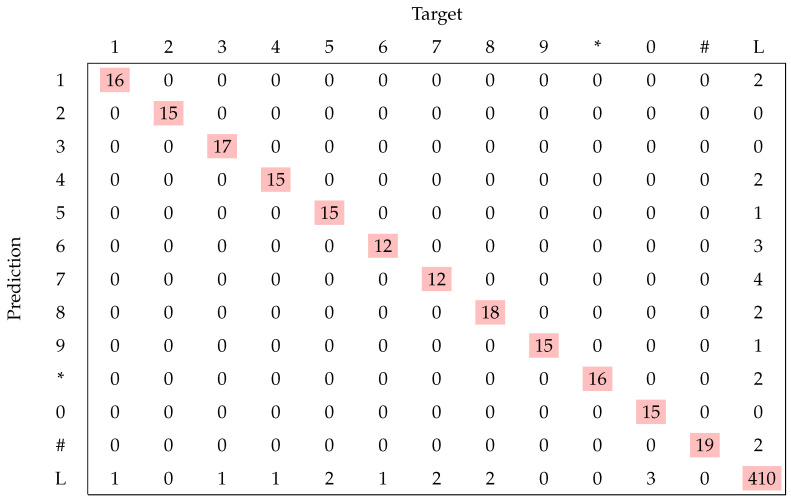
Confusion matrix with the access control keypad.

**Figure 10 sensors-22-03183-f010:**
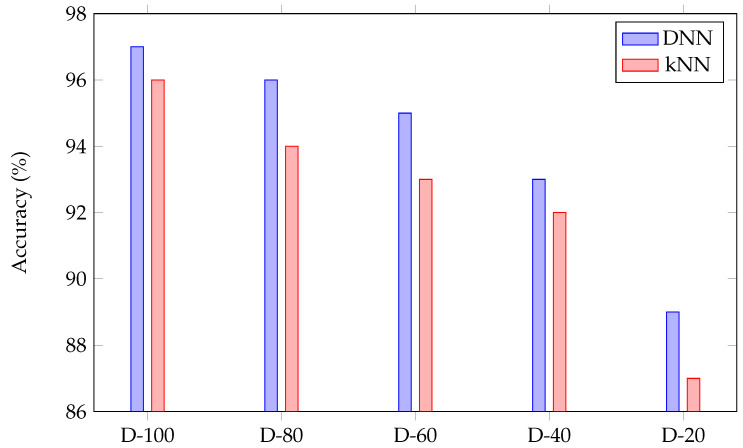
Comparison of accuracy between DNN and kNN approaches considering different data size, N% of the data (D-N) for the access control keypad.

**Figure 11 sensors-22-03183-f011:**
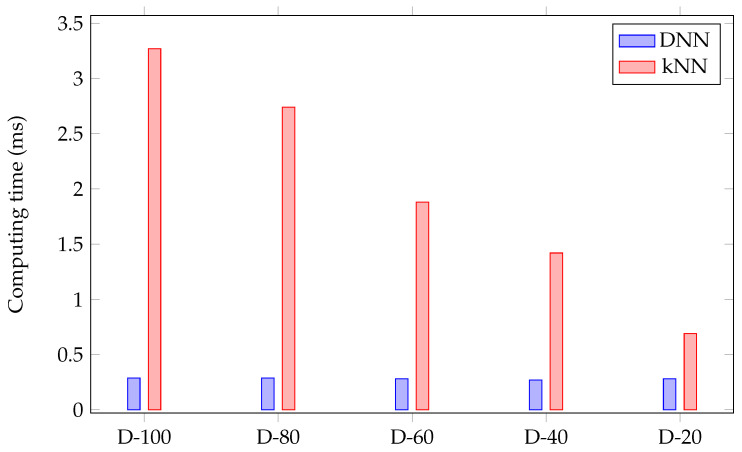
Comparison of computing time between DNN and kNN approaches considering different data size, N% of the data (D-N) for the access control keypad.

**Figure 12 sensors-22-03183-f012:**
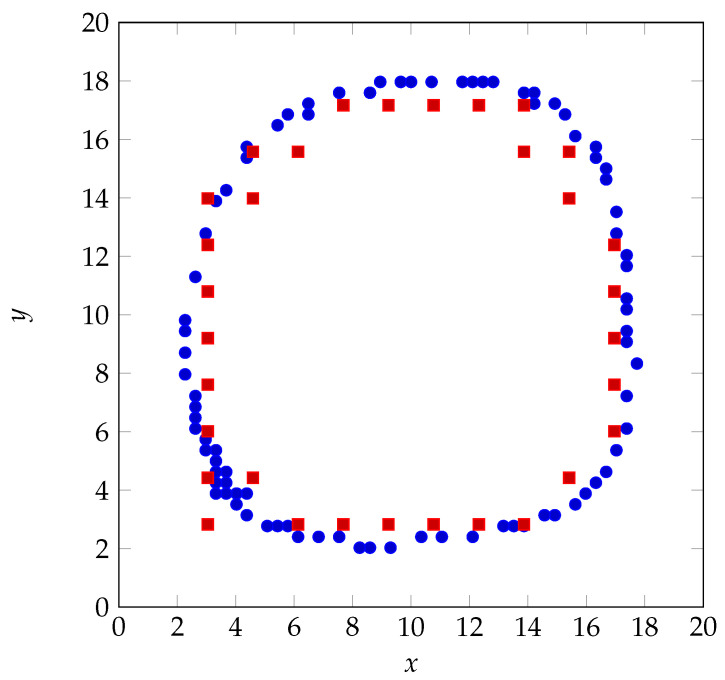
Human finger localization when using classification (C-10): baseline (blue) vs. prediction (red).

**Figure 13 sensors-22-03183-f013:**
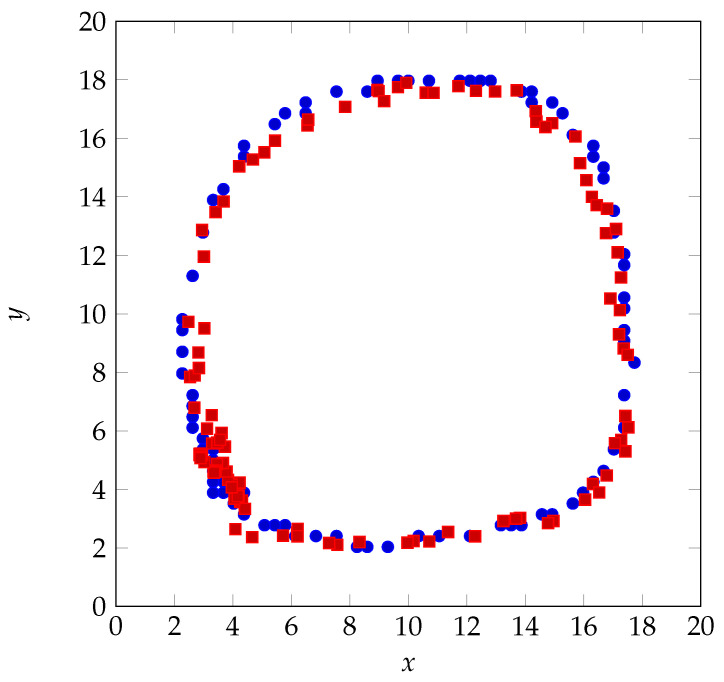
Human finger localization when using regression (R): baseline (blue) vs. prediction (red).

**Figure 14 sensors-22-03183-f014:**
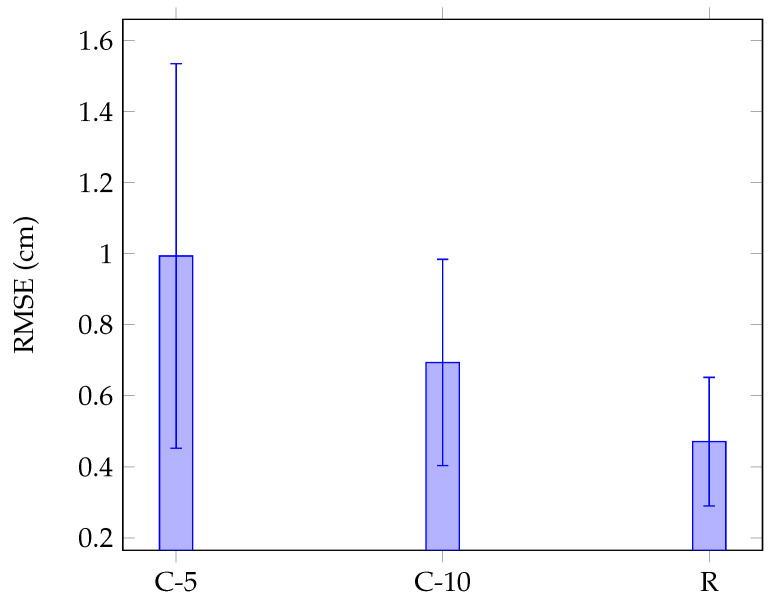
Comparison of human finger localization RMSE (cm) between classification with grid resolution *N* (C-*N*), and regression (R) in the frequency domain.

**Table 1 sensors-22-03183-t001:** Configurations used for touch zone classification.

Grid Resolution (*N*)	Number of Classes ($N^{2}$)	Class Area (cm^2^)
2	4	10.0
3	9	4.4
4	16	2.5
5	25	1.6
6	36	1.1
7	49	0.8
8	64	0.6
9	81	0.5
10	100	0.4

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
