# Peer review of "Machine Learning for Touch Localization on an Ultrasonic Lamb Wave Touchscreen"

_sensors, 2022, doi:10.3390/s22093183_

Round 1

Reviewer 1 Report

Dear Authors,  

   Thanks so much for your manuscript submission to special issue on MDPI Journal of Sensors. This paper proposed a simple deep neural network (DNN) based classification and regression for the performance of touch localization on a tactile surface when applying ultrasonic guided waves. The localization results were presented for human-machine interactions, and the classification scheme was reported with 97% accuracy of identify touches on an access control keypad layout. The authors concluded that applying their DNN-based approaches, accurate and robust touch localization using ultrasonic guided waves can be achieved and stand for viable replacement for signal processing based methods.

    This is a short research article, while the organization of this paper needs some adjustment, the use of English on this research article is acceptable, in spite that some subsections need improvement. Hence, I think that some edits to improve the major and minor problematic issues are necessary, after addressing the comments as suggested, it might be recommented for further acceptance with minor edits.

   Major problematic issues suggested to be addressed are listed as below:

   a) Abstract session: The current version is quite short and generic. I would recommend to extend this paragraph into 120~150 words, which should look more professional. Also, description of your scheme (2nd sentence) is not clear. Please clarify and be more specific. Also, keynote concluding remarks on quantitative results, are expected to be strengthened.

   b) Introduction: this section cited 20+ references and challenged the problems on the technical issues then presented a brief summary on the progress of touch screen technology, development of artificial neural network (ANN) and machine learning (ML) based approaches. There is a potential defect that the authors almost cited nothing on deep learning based methods. I suggest the authors to apply required edits to make this section in a coherent way and smooth the use of English (similar edits can be applied to a second section on Related Work, if inserting that part). In addition, this section fails to present a short summary on major contribution of their work (2-4 manifolds). The organization on the rest of this paper should be shifted to a separate paragraph. Hence, please consider doing a major rewrite on the last 2-3 paragraphs and be more specific.

   c) Approach (Sections 2-4): the schematic flowchart of experimental setup was depicted in Fig. 1. Kernel methods of feature extraction and classification were presented in Fig. 3 and Fig. 5, respectively. The arrangement of images are fine, the resolution and quality of each figure are acceptable. This part has the following problems: it lacks any derivation of mathematical modeling, and performance metrics are very few except for RMSE. Also, I think the quantitative results are quite limited, which may undermine the convincibility of your DNN based approach. Have you considered any idea to fill in this defect? In addition, do you have another option to emphasize the classification accuracy of 97%?  Some statements need adjustment right after the title of this figure in brief descriptions.  

  d) Section 5 (Applications): The titles of Figs. 8-9 need middle alignment. I think it is better to present some discussions within or after the visual results. A subset of their approach which missed any mathematical modeling, the limitations of your study,  or at least a brief summary on your approach, are advocated. If possible, please add a paragraph to explain that issue or at least enclose the explantations in the revision report. Thanks very much!  

   e) Additional comments on Figures: Figs. 7, 10-12, I think the values at y-axis are too sparse. The updated version should use better step size on RMSE, i.e., replace (0.5 1 1.5) with (0.2 0.4 0.6 ... 1.6). If you have better ideas, I would recommend the authors to prepare alternative plans to present the visual and quantitative performance. With respect to the metrics, possible ablation study, sensitivity analysis in tabulated styles, are preferable on acceptance.

   f) Conclusions: this section lacks specific details on limitations of your study, and the last sentence on further study looks generic. The former part needs some supplemental work, and the latter part requires a rewrite (and it is better to be presented in a separate paragraph at the end of this section).

   g) References: Citation formats for each references should be updated with respect to the MDPI template on Sensors Journal. Abbreviated terms on the title of journal names and original styles on citing conference proceedings (including the dates and locations) need to be posted. Meanwhile, I think the authors may consider adding more state-of-the-art publications in Years 2018-2019, 2021-2022, especially some related MDPI publications to the References. Besides, I think the deep learning based models (not just DNN, any weakly or semi-supervised learning schemes) should be strengthened with respect to the updated introduction part.

   Some minor problematic issues which may require further calibrations:

   a) The use of English is acceptable in most of the parts, please check some phrases and sentences which might not be problem free, i.e., Lines 123-124, "vectors of 49 elements each"    

   b) Apply the same (or similar) font style on the legends and marks of each figure (and subdiagrams), be sure the size and resolution are fine enough for publication standard. Rearrange the step size of number in y-axis in several subfigures. 

   c) Step hyphenating a word that may cross over two adjacent lines (which can be adjusted by applying edits to the MS word template)..

   d) Proofreading is required for the whole context, especially in some long paragraphs in the Sections 3-4.     

   Once again, thank you, and best of luck to your further edits. We look forward to reviewing your updated manuscript coming into final acceptance!

Best wishes,

Yours sincerely,

Reviewer 2 Report

The paper is well organized and demonstrates an excellent work on the machine learning-based Lamb wave touchscreen. The paper can be accepted after minor revision (mandatory):

1. In the title, the term of “Ultrasonic Wave” is not rigorous, please revise it as “Ultrasonic Lamb wave” or “Lamb wave”.

2. In the part of 3. Database, the description about the experiment detail is not clear. Why the position of the contacts is randomly selected and how to calculate the number of contact as 6404? It is hard to understand the exact waveform and parameters of the excitation signal from the sentence of “a swipe excitation signal of 1 ms duration from 50 kHz to 100 kHz is generated with a sampling frequency of 500 kHz”. Please give the exact waveform in the Fig.3.

3. 10 and Fig.11 only demonstrate the localization results of the contacts that are not in the central area of the screen. Please provide results of localization error for the contacts in the central area of the screen.
